# Mutations in *hik26* and *slr1916* lead to high-light stress tolerance in *Synechocystis* sp. PCC6803

Katsunori Yoshikawa[1,3], Kenichi Ogawa[1,3], Yoshihiro Toya[1,3], Seiji Akimoto [2], Fumio Matsuda[1] & Hiroshi Shimizu [1✉]

Increased tolerance to light stress in cyanobacteria is a desirable feature for their applications. Here, we obtained a high light tolerant (Tol) strain of *Synechocystis* sp. PCC6803 through an adaptive laboratory evolution, in which the cells were repeatedly sub-cultured for 52 days under high light stress conditions (7000 to 9000 µmol m$^{-2}$ s$^{-1}$). Although the growth of the parental strain almost stopped when exposed to 9000 µmol m$^{-2}$ s$^{-1}$, no growth inhibition was observed in the Tol strain. Excitation-energy flow was affected because of photosystem II damage in the parental strain under high light conditions, whereas the damage was alleviated and normal energy flow was maintained in the Tol strain. The transcriptome data indicated an increase in *isiA* expression in the Tol strain under high light conditions. Whole genome sequence analysis and reverse engineering revealed two mutations in *hik26* and *slr1916* involved in high light stress tolerance in the Tol strain.

[1] Department of Bioinformatic Engineering, Graduate School of Information Science and Technology, Osaka University, Suita, Osaka, Japan. [2] Department of Chemistry, Graduate School of Science, Kobe University, Kobe, Hyogo, Japan. [3]These authors contributed equally: Katsunori Yoshikawa, Kenichi Ogawa, Yoshihiro Toya. ✉email: shimizu@ist.osaka-u.ac.jp

Cyanobacteria are promising host microorganisms for the production of industrial bio-molecules because these cells can directly fix carbon dioxide through the Calvin Benson cycle and converted it into target compounds using light energy[1]. However, excess light, such as in summer, damages their photosystem and inhibits photosynthetic activity[2]. Because high light (HL) stress is one of the major obstacles in the bio-production of cyanobacteria, it is highly desired to develop an HL tolerant strain for enhancing their growth and production rate of various bio-molecules for industrial applications.

Cyanobacteria can adapt their cellular system to survive under HL stress conditions by reducing their light-harvesting antenna size[3] and decreasing contents of chlorophyll and photosystem I (PSI)[4–6]. Furthermore, they possess some mechanisms for preventing photoinhibition by allowing excess light energy to escape, such as state transition[7], heat dissipation[8], energy dissipation from phycobilisome and photosystem II (PSII)[9,10], and synthesis and maintenance of protein complexes in thylakoid membrane by chaperone proteins[11]. However, because these mechanisms are insufficient for protecting the cellular system from excess light energy under further strong light intensity conditions (more than 2000 μmol m$^{-2}$ s$^{-1}$), the cell growth is reduced due to inhibition of photosynthetic activity caused by damage to their photosystems. It has been reported that degradation of the PSII reaction center[12] and inhibition of protein synthesis and repair of PSII protein[13] by reactive oxygen species occur as by-products of photosynthesis. Although some mutants with reduced light-harvesting pigment content and minimized light-harvesting antenna size have been developed in cyanobacteria for improving the HL stress tolerance in previous studies, such strains have certain drawbacks that cause growth defects under low light conditions[14,15]. Furthermore, transcriptome analysis revealed a downregulation of genes related to light-harvesting systems, flagellum synthesis, and transhydrogenase, and upregulation of D1 protein turnover-related genes under excessive HL conditions[16]. However, effective mechanisms for protecting cells from excess light without decreasing the antenna size are still unknown.

Adaptive laboratory evolution (ALE) experiment is a method used to induce tolerance against stress conditions[17]. A serially passaged culture is performed under a stress environment with decreasing the growth rate, which allows the selection of mutants with improved growth phenotypes. Compared to random mutagenesis approaches using mutagens, this approach can be easily used for identifying key mutations to improve the evolved strains by genome sequence analysis due to their low mutation frequency[18,19]. Furthermore, a comparison between the parental and evolved strains employing "Omics" technologies provides useful clues for understanding the tolerance mechanisms. Many tolerant strains of cyanobacteria against various stress conditions, such as alcohol, acid, and heavy metals, have been developed[20–22]. In the present study, an evolved strain of *Synechocystis* sp. PCC6803 (hereafter referred to as PCC6803) was obtained through an ALE experiment under HL stress conditions. We identified key mutations involved in the HL tolerance using whole-genome sequencing and reverse engineering. We also aimed to reveal HL tolerance mechanisms in the evolved strain by fluorescence analyses of their photosystem as well as by transcriptome analysis.

## Results

**ALE experiment under high light condition**. To analyze the effect of HL stress on growth, the PCC6803 strain was cultured under several light intensities (40, 4000, 7000, and 9000 μmol m$^{-2}$ s$^{-1}$) with bubbling of air. Although cell growth was not inhibited under light conditions of 4000 μmol m$^{-2}$ s$^{-1}$, it was decreased under 7000 μmol m$^{-2}$ s$^{-1}$ and almost stopped under 9000 μmol m$^{-2}$ s$^{-1}$ (Fig. 1).

ALE experiment was performed in four parallel test tubes. After growing the cells, the culture with the highest cell concentration was inoculated into fresh medium in four test tubes (Fig. 2a). Because the growth rate was gradually increased during the experiment, the light intensity was initiated at 7000 μmol m$^{-2}$ s$^{-1}$ and increased stepwise to 9000 μmol m$^{-2}$ s$^{-1}$. Specific growth rate and light intensity during the evolution process are shown in Fig. 2b. After subculturing the cells 23 times, the evolved cells were grown under 9000 μmol m$^{-2}$ s$^{-1}$, wherein the PCC6803 strain showed severe growth inhibition; these cells were named as "Tol strain" and stored at −80 °C. The Tol strain showed higher HL tolerance compared with PCC6803, i.e., the growth of the Tol strain did not decrease under 7000 μmol m$^{-2}$ s$^{-1}$ and even under 9000 μmol m$^{-2}$ s$^{-1}$ (Fig. 1). Interestingly, the growth curve of the Tol strain was identical to that of PCC6803 strain under low light condition (40 μmol m$^{-2}$ s$^{-1}$), suggesting the Tol strain could adapt its photosystem or metabolism to various light intensities (Fig. 1). The stability of the HL tolerance of the Tol strain under 9000 μmol m$^{-2}$ s$^{-1}$ was confirmed after serial transfer cultivations under low light intensity (40 μmol m$^{-2}$ s$^{-1}$) for 15 days following five subcultures (Supplementary Fig. 1). This result indicates that the HL tolerance was not caused by acclimation to the HL stress condition during the ALE experiment, but probably by the introduction of mutations in the genome of the Tol strain.

As there was a possibility that the Tol strain was a mixture of different phenotype or genotype strains, four single colonies of the Tol strain were isolated and named Tol(S1), Tol(S2), Tol(S3), and Tol(S4). All the four strains showed identical growth curves as that of the Tol stain under both low light and HL conditions (Supplementary Fig. 2). These results suggest that the population in the Tol strain was entirely composed of the strain possessing HL tolerance and high growth under low light conditions.

**High light tolerant mechanism of Tol strain**. The photosynthesis ability of the Tol strain obtained by the ALE experiment was investigated. Chlorophyll is an important pigment for light harvesting during photosynthesis. It has been reported that the amount of chlorophyll decreases to avoid absorption of excess light under HL conditions[23]. The chlorophyll contents of the PCC6803 and Tol strains under various light conditions are shown in Fig. 3. In the PCC6803 strain, chlorophyll content under light conditions of 7000 and 9000 μmol m$^{-2}$ s$^{-1}$ was 52% and 64% lower, respectively, than that under 4000 μmol m$^{-2}$ s$^{-1}$, which was consistent with a previous report[23]. In the Tol strain, chlorophyll content was only decreased by 21% and 42% under 7000 and 9000 μmol m$^{-2}$ s$^{-1}$, respectively. These results suggest that the Tol strain possesses a mechanism for HL tolerance other than decreasing the chlorophyll content.

In the cultures of PCC6803 and Tol strains under each light condition, absorption spectra of the cells were analyzed at 48 h (Fig. 4a). The phycobilisome and chlorophyll exhibited absorption bands around 620 and 675 nm, respectively. Although the relative content of chlorophyll to phycobilisome decreased as the light intensity increased in the PCC6803 strain, it was stable in the Tol strain. The comparison of absorption spectra suggests the relative carotenoid content in the Tol strain is decreased compared to PCC6803 strain. The relative intensities of PSII fluorescence (680–700 nm) under light conditions of 4000 and 7000 μmol m$^{-2}$ s$^{-1}$ were almost identical in the PCC6803 strain (Fig. 4b), whereas it increased under 9000 μmol m$^{-2}$ s$^{-1}$, thereby suggesting a damage to PSII (Fig. 4c). By contrast, an increase in PSII fluorescence was not observed in response to a change from

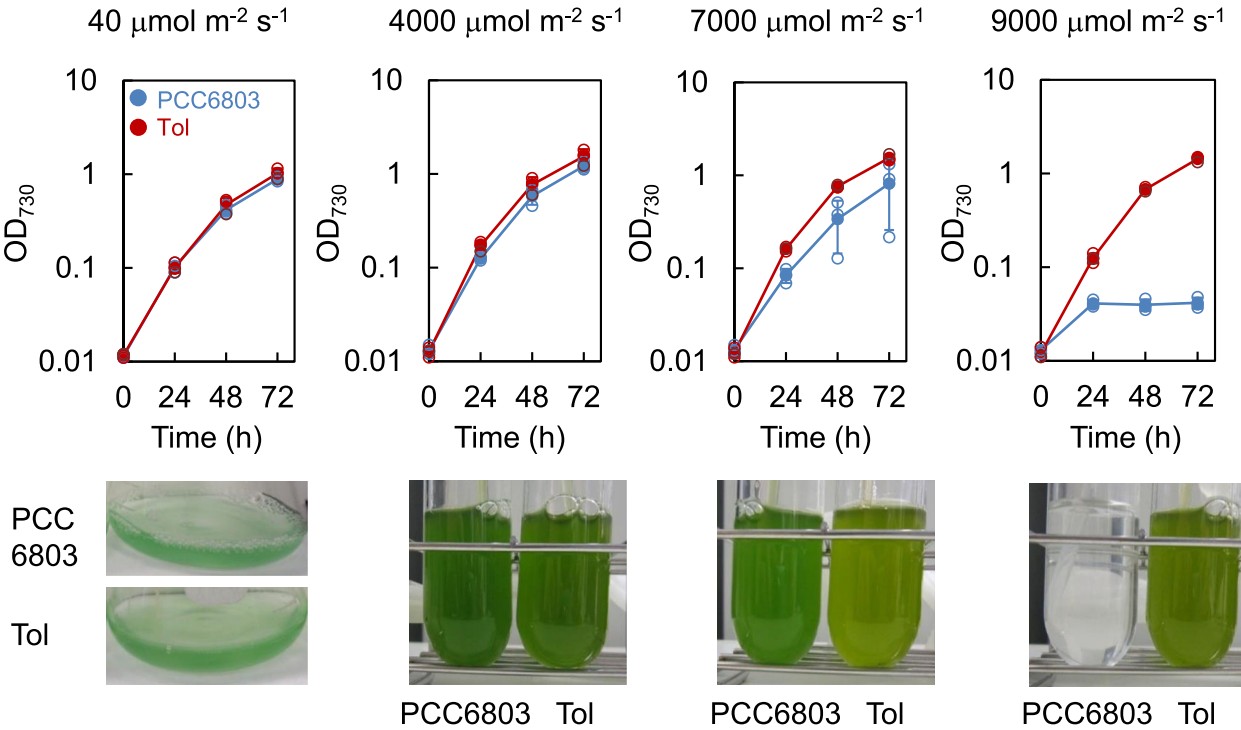

**Fig. 1 Cell growth of PCC6803 and Tol strains under different light intensity conditions.** Cultures under 40 µmol m$^{-2}$ s$^{-1}$ were performed using an LED plate, and cultures under other light intensities (4000, 7000, and 9000 µmol m$^{-2}$ s$^{-1}$) were performed using point source LED. All photos of cultures were taken at 72 h. No further growth was observed in the PCC6803 strain after 72 h under the 9000 µmol m$^{-2}$ s$^{-1}$ (Supplementary Fig. 1). The filled blue and red circles indicate average of OD$_{730}$ measurements of PCC6803 and Tol strains, respectively. Error bars indicate standard deviation from triplicate experiments. The open symbols represent individual data points in the triplicate experiments.

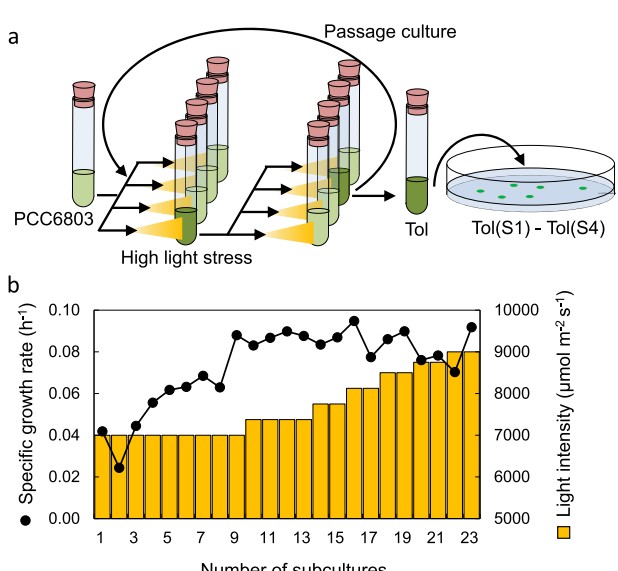

**Fig. 2 ALE experiment under high light stress condition. a** Experimental design of the ALE experiment. Cells were cultured in a test tube using a point source LED. Among four parallel cultures, the culture broth with the highest growth rate was inoculated into a fresh medium every 2–3 days. **b** Changes in specific growth rate during ALE experiment. Line plot indicates specific growth rate and bars indicate light intensity at each subculture experiment.

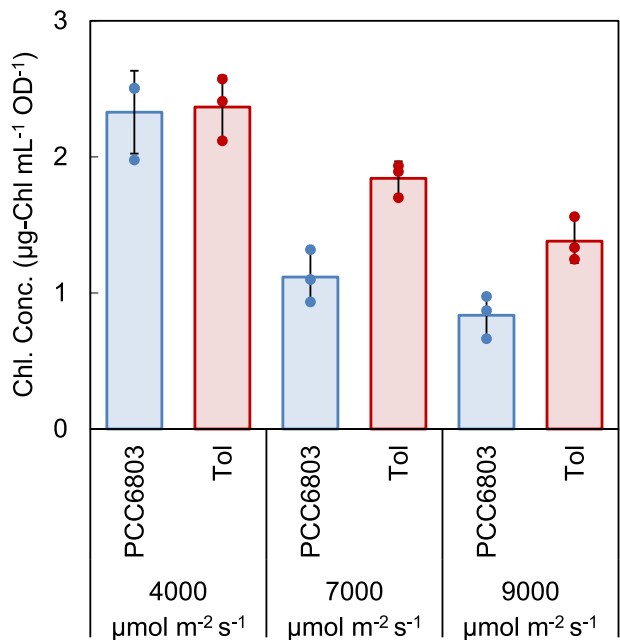

**Fig. 3 Chlorophyll content of PCC6803 and Tol strains under different light intensities.** The chlorophyll contents were measured at 48 h for all conditions. The bars and plots represent averages and individual data points in the biological triplicate experiments. Error bars indicate standard deviation from the triplicate experiments.

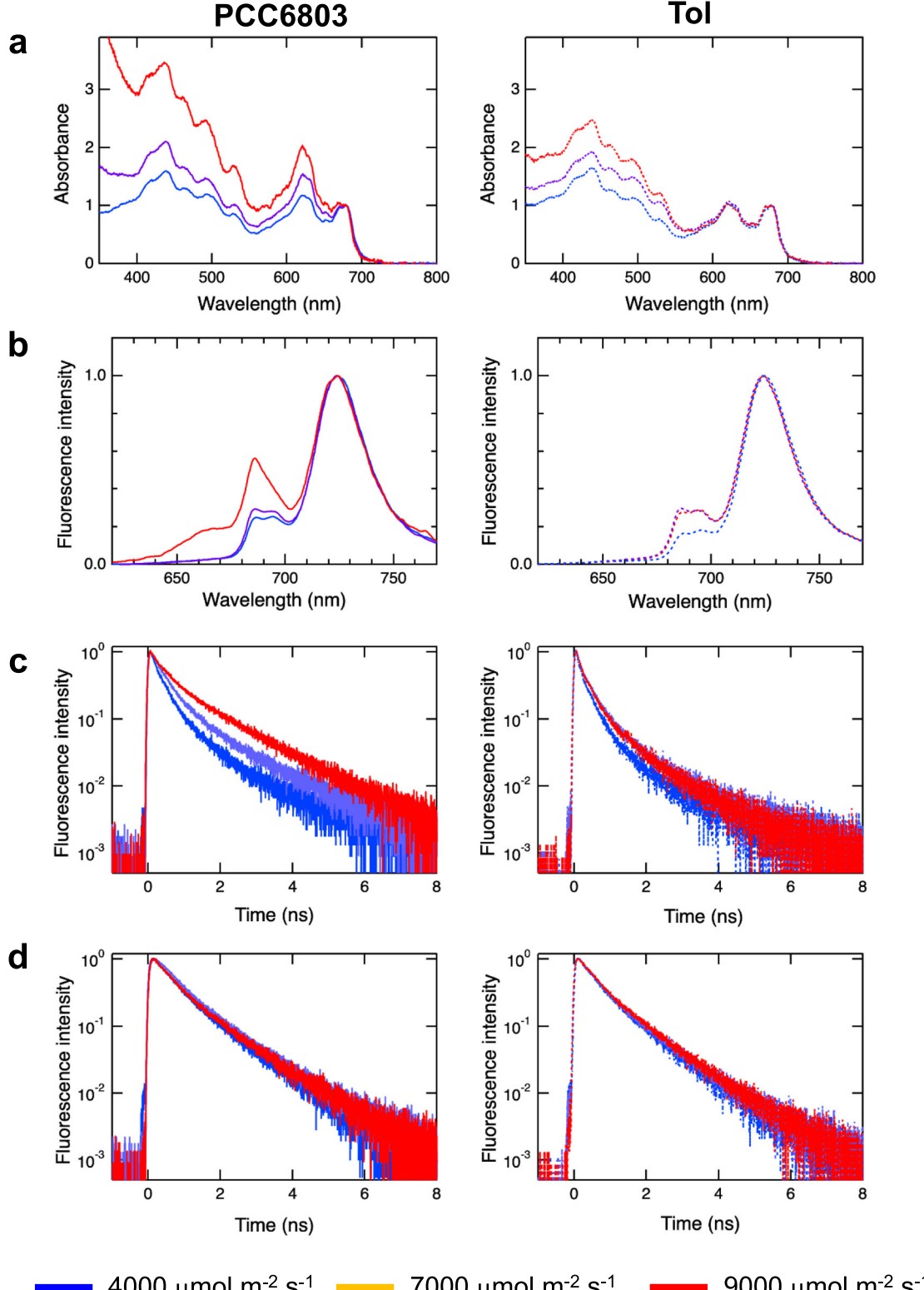

**Fig. 4 Characterization of photosynthetic abilities in PCC6803 and Tol strains. a** Absorption spectra normalized at the chlorophyll Qy band.
**b** Fluorescence spectra normalized at the PSI band (<725 nm). **c** Fluorescence decay curves of PSII normalized to maximum intensities. **d** Fluorescence decay curves of PSI normalized to the maximum intensities. Graphs to the left and right represent PCC6803 and Tol strains, respectively. Blue, purple, and red lines represent 4000, 7000, and 9000 $\mu mol\ m^{-2}\ s^{-1}$, respectively. The absorption spectra, fluorescence spectra, and fluorescence decay curves were measured at 48 h for all conditions.

7000 to 9000 µmol m$^{-2}$ s$^{-1}$ for the Tol strain. This result suggests that the ratio between PSII and PSI was maintained in the Tol strain under HL conditions.

As PSII in the Tol strain was observed to be stable under HL conditions (Fig. 4b), the excitation-energy flows in PSII and PSI were analyzed by time-resolved fluorescence spectroscopy (Fig. 4c, d). In the PCC6803 strain, a decrease in fluorescence decay rate in PSII was observed as the light intensity increased, suggesting an impairment of PSII. On the contrary, fluorescence decay curves were almost identical in the Tol strain regardless of the light intensity, suggesting the normal energy flow in PSII under HL conditions. However, a decrease in the fluorescence decay rate was not observed in PSI under HL conditions in both PCC6803 and Tol strains. These findings suggest that the growth inhibition of the PCC6803 strain under HL conditions was caused by a change in energy flow due to PSII damage. As the damage was alleviated and energy flow was maintained at normal state in the Tol strain, the cells could remain stable under HL condition without growth inhibition.

**Functional analysis of transcriptome data**. Transcriptome analysis was performed using a DNA microarray to understand the HL tolerance mechanisms in the Tol strain. Four sets of microarray data were obtained for the PCC6803 and Tol strains grown under 4000 and 7000 µmol m$^{-2}$ s$^{-1}$. The microarray data of the PCC6803 and Tol strains under 4000 and 7000 µmol m$^{-2}$ s$^{-1}$ were designated as W-4000 and W-7000, and T-4000 and T-7000, respectively. Gene expression data for 2942 of 3527 genes spotted on the microarray were obtained in all four conditions (Supplementary Data 1).

Table 1 summarizes the number of genes showing significant difference in gene expression levels between the two conditions. Differences were compared with the two-tailed Student's $t$ test. To correct the $p$ values for multiple comparisons, the $q$ values were calculated with considering false discoveries by Storey's method. When the expression data between 4000 and 7000 µmol m$^{-2}$ s$^{-1}$ were compared for each condition, only a small number of genes showed different expression levels (0.1% between W-4000 and W-7000; 0.1% between T-4000 and T-7000). This result suggested that the 4000 µmol m$^{-2}$ s$^{-1}$ condition for the Tol strain is also an HL condition for PCC6803. In the comparison between W-7000 and T-7000, the expression levels of 28% genes shows significant difference between PCC6803 and Tol strains under 7000 µmol m$^{-2}$ s$^{-1}$.

To identify genes whose expression was significantly altered in the Tol strain compared to those in PCC6803 strain, the ratios of gene expression level were ranked. Table 2 shows the top 10 genes of increased or decreased expression levels in the Tol strain under HL condition. Among these candidates, the expression ratios of *isiA* and *isiB* were dramatically increased in the Tol strain (27-fold in *isiA* and 9.5-fold in *isiB*). The transcriptome data in a volcano plot for comparing T-7000 and W-7000 are shown in

Supplementary Fig. 3. The *isiA* and *isiB* encode iron-stress chlorophyll-binding protein (CP43′) and flavodoxin, respectively. It has been reported that *isiA* deletion causes complete destruction of photosynthetic function under HL condition, whereas *isiB* deletion is not related to HL tolerance[24].

**Effect of *isiA* overexpression on HL tolerance**. Since the *isiA* expression level in T-7000 was higher than that in T-4000 and W-7000, the enhanced expression of *isiA* might have been involved in HL tolerance in the Tol strain. To confirm this hypothesis, an *isiA* overexpressed strain of PCC6803, named PCC6803/OE-isiA, was constructed by introduction of *isiA*, controlled under a strong promoter of *psbA2*, to the neutral site of PCC6803. The PCC6803/OE-isiA showed increased growth under HL conditions (7000 and 9000 µmol m$^{-2}$ s$^{-1}$) compared with its parent strain PCC6803 (Fig. 5d). An increase in PSII fluorescence due to *isiA* overexpression has been reported[25], which is consistent with the results of chlorophyll fluorescence and fluorescence decay curve in the Tol strain under HL condition in our study.

However, the HL tolerance of the PCC6803/OE-isiA was lower than that of the Tol strain (Fig. 5b), suggesting there are other mechanisms contributing toward HL tolerance of the Tol strain. *isiA* encodes CP43′, which is required for the formation of PSI supercomplex and efficient state transition[26] and is necessary for HL tolerance[24]. Consequently, in our study, higher expression of *isiA* in the Tol strain may have enhanced non-photochemical quenching and efficient electron transfer under HL condition, thus increasing HL tolerance. Furthermore, because the *isiA-isiB-sll0249* is an operon on the genome, the expressions of *isiB* and *sll0249* as well as *isiA* were increased in the Tol strain (Table 2). Therefore, overexpression of these three genes may lead to stronger HL tolerance.

**Whole-genome sequence of the HL tolerant strain**. To identify mutations in the Tol strain, whole-genome sequences of PCC6803 and Tol strains were analyzed by a MiSeq Illumina sequencer using a 150 bp paired-end method. The mapping results that were compared to a reference genome sequence of *Synechocystis* sp. PCC6803 GT-I strain (NCBI Reference Sequence No.: NC_017038.1)[27] showed around 100% coverage of the whole-genome sequence and average 500 read depth for each nucleotide sequence in both the strains (Supplementary Table 1). In the PCC6803 strain, seven mutations were identified compared with the reference genome sequence. In the Tol strain, additional two non-synonymous mutations were introduced in *hik26* and *slr1916* (Supplementary Table 2). *slr1916* is annotated as a probable esterase, and its deletion was reported to cause increased PSI content under HL condition[28]. *hik26* encodes a two-component sensor histidine kinase, but its responsive stress conditions and response regulators have not been identified.

**Effects of the *hik26* and *slr1916* mutations on HL tolerance**. Effects of the identified mutations on HL tolerance were analyzed by deletion of *hik26* or *slr1916* in PCC6803 and Tol(S1) strains, and introduction of mutated *hik26* or *slr1916* into PCC6803.

The *slr1916* was deleted in both the PCC6803 and Tol(S1), generating 6803Δslr1916 and Tol(S1)Δslr1916, respectively. The 6803Δslr1916 strain could grow under 9000 µmol m$^{-2}$ s$^{-1}$ similar to the Tol strain and its culture broth showed higher viscosity (Fig. 5e). The Tol(S1)Δslr1916 strain maintained HL tolerance but its broth did not show high viscosity (Fig. 5f). These results indicate the possibility of the mutation of *slr1916* in the Tol strain being a loss-of-function mutation, which may have increased HL tolerance. However, the 6803Δslr1916 and Tol strains showed different phenotypes, such as high viscosity and cell aggregation.

**Table 1 Number of genes showing significant difference in expression levels.**

|  | W-4000 | W-7000 | T-4000 | T-7000 |
|---|---|---|---|---|
| W-4000 | – | 2 (0.1%) | 13 (0.4%) | 455 (15.5%) |
| W-7000 | – | – | 147 (5.0%) | 824 (28.0%) |
| T-4000 | – | – | – | 2 (0.1%) |
| T-7000 | – | – | – | – |

Numbers indicate the number of genes whose expression levels were significantly different between the two conditions among 2942 genes ($q$ value < 0.05). The numbers in parenthesis indicate the percentage of the identified genes of the 2942 genes.

**Table 2 Top 10 genes of increased or decreased expression levels with significant $q$ values in Tol strain under HL condition ($7000\ \mu mol\ m^{-2}\ s^{-1}$).**

| ORF ID | Gene name | Function | Functional category | Expression ratio[a] | $q$ value |
|---|---|---|---|---|---|
| **Increased expression** | | | | | |
| *sll0247* | *isiA* | Iron-stress chlorophyll-binding protein | Photosystem II | 27 | 0.012 |
| *sll0248* | *isiB* | Flavodoxin | Soluble electron carriers | 9.5 | 0.024 |
| *ssl2920* | – | Hypothetical protein | | 7.7 | 0.014 |
| *slr1417* | *ycf57, iscA* | Hypothetical protein | | 7.5 | 0.010 |
| *sll1054* | – | Hypothetical protein | | 7.3 | 0.007 |
| *ssr2194* | – | Unknown protein | | 6.9 | 0.044 |
| *sll0249* | – | Hypothetical protein | | 6.3 | 0.018 |
| *slr5055* | – | Similar to UDP-N-acetyl-D-mannosaminuronic acid transferase | | 5.6 | 0.049 |
| *slr1913* | – | Hypothetical protein | | 5.3 | 0.023 |
| *slr1282* | ISY508b | Putative transposase | Transposon-related functions | 5.2 | 0.013 |
| **Decreased expression** | | | | | |
| *slr2076* | *groEL1, cpn60-1* | Chaperonin | Chaperones | 0.18 | 0.010 |
| *slr1963* | – | Water-soluble carotenoid protein | | 0.19 | 0.013 |
| *slr0967* | – | Hypothetical protein | | 0.20 | 0.044 |
| *ssl3769* | – | Unknown protein | | 0.20 | 0.013 |
| *ssr1155* | – | Hypothetical protein | | 0.22 | 0.036 |
| *slr0447* | *urtA, amiC* | ABC-type urea transport system | | 0.23 | 0.018 |
| *sll1745* | *rplJ, rpl10* | 50S ribosomal protein L10 | | 0.23 | 0.011 |
| *sll1322* | *atpI, atpB* | ATP synthase A chain of CF(0) | | 0.23 | 0.010 |
| *sll0170* | *dnaK2, dnaK* | DnaK protein 2, heat shock protein | Chaperones | 0.23 | 0.018 |
| *sll0226* | *ycf4* | Photosystem I assembly related protein | Photosystem I | 0.24 | 0.009 |

[a]Geometric mean of triplicate experiments under each condition was used for calculating the ratio of gene expression level in Tol strain to that in PCC6803 strain.

The different phenotypes between the Tol(S1)Δslr1916 and the 6803Δslr1916 strains may have been caused due to synergy effects of the *slr1916* deletion and *hik26* mutation.

The deletion strains of *hik26* in PCC6803 and Tol(S1), designated as 6803Δhik26 and Tol(S1)Δhik26, showed severe growth inhibition under HL conditions and similar growth rate as that of with PCC6803 strain under low light condition ($40\ \mu mol\ m^{-2}\ s^{-1}$) (Fig. 5h, i). This indicates that *hik26* is specifically needed for HL tolerance. To analyze the effect of the mutation of *hik26* on HL tolerance, mutated *hik26* (hereafter called as hik26m) with its own promoter region was amplified from the Tol(S1) strain and introduced into the genome of the 6803Δhik26, generating 6803Δhik26/hik26m. The 6803Δhik26/hik26m showed identical growth as that of the Tol strain under all conditions, even under $9000\ \mu mol\ m^{-2}\ s^{-1}$ (Fig. 5j); however, high viscosity and cell aggregation were not observed in the 6803Δhik26/hik26m. Both the *hik26* mutation and *slr1916* deletion also produced HL tolerance. Therefore, it seems that the mutations in either gene can cause HL tolerance, and the phenotype of the Tol strain could be attributed to either or both. *hik26* encodes a histidine kinase, which is a component of two-component sensor-transducer system, but environmental conditions that activate Hik26 and its response regulator have not been identified. A deletion of *hik26* caused severe growth inhibition under HL conditions, and introduction of hik26m improved HL tolerance, thus suggesting that Hik26 probably senses HL and regulates the genes required for HL tolerance through a response regulator that has not yet been identified. As well as *hik26*, mutated *slr1916* (hereafter called as slr1916m) with its own promoter region was amplified from the Tol (S1) strain and introduced into the genome of the 6803Δslr1916, generating 6803Δslr1916/slr1916m. The 6803Δslr1916/slr1916m showed almost identical growth as that of the Tol and the 6803Δslr1916 strains under all conditions (Fig. 5g). Furthermore, the culture broth of PCC6803Δslr1916/slr1916m showed high

viscosity and cell aggregation under the HL conditions, as well as PCC6803Δslr1916. These results suggest that the mutation of *slr1916* in the Tol strain is loss-of-function mutation.

## Discussion

In the present study, an HL stress tolerant (Tol) strain of *Synechocystis* sp. PCC6803 was obtained by an ALE experiment under extreme HL stress condition ($9000\ \mu mol\ m^{-2}\ s^{-1}$). A previous study has reported that size reduction of the light-harvesting antenna increased HL tolerance but decreased the growth under low light condition[29]. However, we observed that the growth rate of the newly obtained Tol strain was identical to that of the PCC6803 strain under low light condition ($40\ \mu mol\ m^{-2}\ s^{-1}$), suggesting that the Tol strain possesses a different mechanism to overcome HL stress. This is consistent with the findings that the decrease in chlorophyll content due to HL stress was partially alleviated in the Tol strain. Furthermore, evaluation of photosynthesis machineries revealed the HL tolerance mechanisms, which indicated that the PSII complex and the electron flow to PSI were maintained even under HL conditions in the Tol strain.

The HL tolerance mechanism in the Tol strain was also investigated based on transcriptome data of PCC6803 and Tol strains and known HL-responsive genes which are commonly induced or repressed under HL conditions[23]. Although a comparison between W-4000 and T-4000 showed that the number of genes whose expression was significantly changed was small (0.4%); a comparison between W-7000 and T-7000 showed that the expression levels of many genes were changed (28.0%). Consequently, a change in expression of these genes would have contributed to the superior HL tolerance in the Tol strain. The transcriptome data revealed that *isiA* and *isiB* expressions were highly induced in the Tol strain under HL condition. The

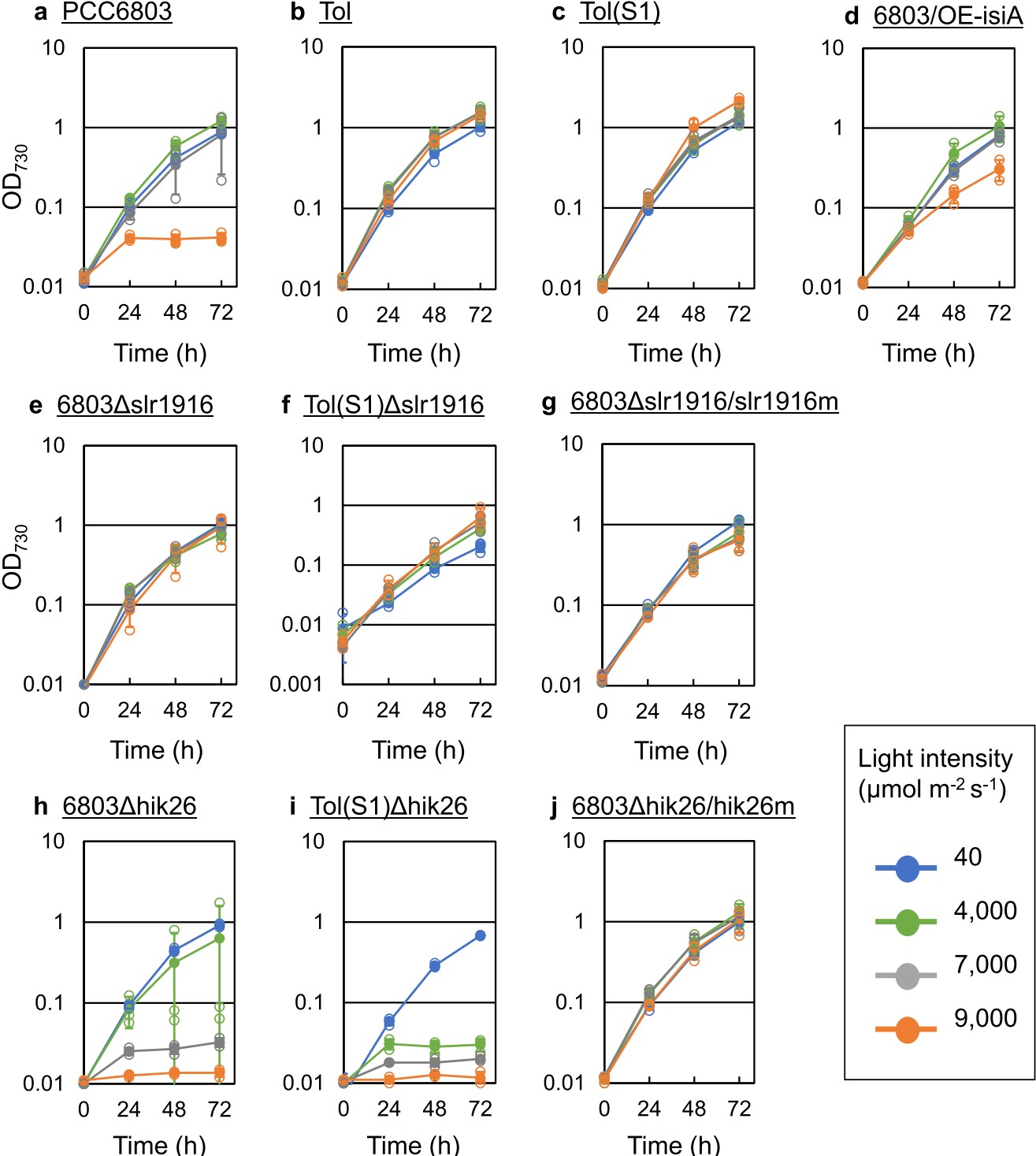

**Fig. 5 Reverse engineering based on transcriptome and whole-genome re-sequencing analyses. a** PCC6803, **b** Tol, **c** Tol(S1), **d** 6803/OE-isiA, **e** 6803Δslr1916, **f** Tol(S1)Δslr1916, **g** 6803Δslr1916/slr1916m, **h** 6803Δhik26, **i** Tol(S1)Δhik26, **j** 6803Δhik26/hik26m strains. Culture under a light intensity of 40 μmol m⁻² s⁻¹ was performed using an LED plate, and cultures under other light intensities (4000, 7000, and 9000 μmol m⁻² s⁻¹) were performed using point source LED. The filled and open circles represent averages and individual data points in the biological triplicate experiments. Error bars indicate standard deviation from triplicate experiments. Note that the errors of the growth curve of the 6803Δhik26 strain under 4000 μmol m⁻² s⁻¹ were large because in the triplicate experiments similar growth as that of PCC6803 was observed in one culture experiment and severe-inhibited growth was displayed in remaining two culture experiments. In the 6803Δslr1916/slr1916m and 6803Δhik26/hik26m strains, the mutated genes were introduced at the neutral sites, not at their natural positions.

*isiA* was originally observed to be an iron deficiency stress-inducible gene[30], and it functioned as an antenna protein for PSI under iron-limiting condition[31]. It has been reported that the *isiA* and *isiB* are induced by various stress conditions. For example, the gene expression is highly induced by hydrogen peroxide, which is generated under an extremely HL condition[32]. The *isiA* deletion inhibits a redistribution of phycobilisome-absorbed energy between PSII and PSI[26] and causes a growth defect under HL condition[33]. Furthermore, the *isiAB* deletion resulted in a photosensitive phenotype under HL condition, with accumulation of reactive oxygen species and cell bleaching[24]. Recently, it has been reported that IsiA is a member of a high light-inducible carotenoid-binding protein complex (HLCC) on a thylakoid membrane, and a lack of HLCC components causes impaired state transition and increased sensitivity to HL stress[34]. The HLCC also plays a role in protecting the thylakoid membrane and D1 protein of PSII from oxidative stress[34]. Therefore, the enhanced expression of IsiA in the Tol strain under HL condition would promote the formation of HLCC, which involves in the state transition for processing excess light energy, thus protecting the thylakoid membrane and D1 protein from oxidative stress.

Two mutations of *slr1916* and *hik26* in the Tol strain were identified by whole-genome sequencing, and the function was evaluated by reverse engineering. Especially, Hik26, a histidine kinase of a two-component system, can regulate many genes through a response regulator. Since the expression level of *hik26* was not significantly altered in all the conditions, the mutation in *hik26* might activate Hik26 at enzymatic level and affect its regulated gene expression level. The non-synonymous mutation in Hik26m (T29K) was located at the sensor domain predicted by NCBI database (ACCESSION No. YP_005652307.1). This suggests that the mutation might activate Hik26 and affect the expression levels of its regulatory genes. Considering functional analysis data and known HL-responsive genes, the known induced HL-responsive genes tended to be highly expressed and repressed HL-responsive genes were poorly expressed in the Tol strain than in PCC6803. The HL tolerance mechanisms related to these genes might have been more activated in the Tol strain, thereby increased HL tolerance. Moreover, these genes may be regulated through Hik26, and the mutation in *hik26* might have caused these expression changes. Homology search using BLASTP (KEGG database) indicated that the homologous protein Hik26 was conserved in many cyanobacteria species except in marine *Synechococcus* and *Prochlorococcus* species, and was not present in eukaryotic photosynthetic organisms such as green algae. The mutation site was highly conserved among cyanobacteria possessing homologous Hik26. Another histidine kinase, Hik33, was also reported to be necessary for HL tolerance, but *hik33* deletion also decreased the growth under low light condition[35]. Hence, our studies suggest that Hik26 is a different histidine kinase which is known so far, specifically responsive to HL condition. Regarding the *slr1916*, recently, comprehensive experiments using an inducible CRISPRi gene repression library have shown that *pmgA* and *slr1916* repression clones exhibit similar growth profiles in response various stresses including light conditions[36]. The PgmA is a regulator involved in HL responses in the PCC6803[33]. A scheme explaining the mechanism of HL tolerance in the Tol strain is summarized in Fig. 6.

In conclusion, we obtained an HL tolerant strain of *Synechocystis* sp. PCC6803 by the ALE and revealed the mechanisms for HL tolerance by absorption and fluorescence analyses of the photosystem. Transcriptome analysis and whole-genome sequence revealed that *isiA* overexpression and *hik26* mutation enhance HL tolerance in the PCC6803 strain. The Hik26 is a different histidine kinase which is known so far, that responds to HL stress. We expect that further extensive studies to identify a response

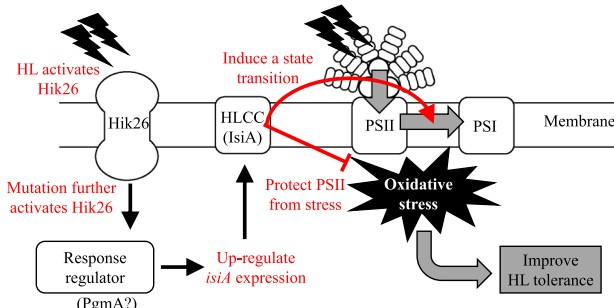

**Fig. 6 A scheme of the mechanism of HL tolerance in the Tol strain.** The Hik26 is a histidine kinase in response to HL stress. The mutation (T29K) of Hik26 in the Tol strain may activate the histidine kinase activity. The Hik26 induces the expression of genes including *isiA* through a response regulator. The IsiA contributes to protect the membrane proteins from the oxidative stress caused by HL conditions, and involves in light state transition from PSII to PSI.

regulator of Hik26 would reveal the whole picture of regulatory mechanisms responsible for HL stress tolerance in the PCC6803 strain.

## Methods

**Strains and culture conditions.** The *Synechocystis* sp. PCC6803 glucose-tolerant, nonmotile[37], GT-I strain was used as the parent strain. A slightly modified BG11 medium was used for the culture experiments[38]. Pre-cultured cells were cultivated in 100 mL Erlenmeyer flasks with 20 mL modified BG11 medium at 34 °C with rotary agitation at 150 rpm (BR-43FL, TAITEC, Japan), under continuous light illumination using a light-emitting diode (LED) plate (about 40 μmol m$^{-2}$ s$^{-1}$; LC-LED 450 W, TAITEC). The pre-cultured cells (OD$_{730}$ = 0.01) were inoculated into fresh modified BG11 medium. The main culture was performed using a test tube ($\varphi$3 × 20 cm, AGC Techno Glass, Shizuoka, Japan) containing 30 mL modified BG11 medium at 34 °C under continuous light illumination at 4000–9000 μmol m$^{-2}$ s$^{-1}$ using a point source LED (LA-HDF158AS, HAYASHI watch-works, Tokyo, Japan). The spectrum of the point source LED is shown in Supplementary Fig. 4. The cultures were aerated by mixing with sterile air. The light intensity (4000–9000 μmol m$^{-2}$ s$^{-1}$) used in the present study was significantly higher compared to that used in other studies. In our previous report[16], the PCC6803 strain exhibited photoinhibition when cultured under 1100–1300 μmol m$^{-2}$ s$^{-1}$. Although a flat LED panel with the photobioreactor was used in the previous study, we here culture the cells in a test tube with a point source LED that can irradiate more HL intensity. The light intensity was measured at the irradiate point on the surface of the test tube. Therefore, the normalized light intensity at the entire test tube becomes low. In the PCC6803 strain, although 70% growth reduction was observed at 1100 μmol m$^{-2}$ s$^{-1}$ with the flat LED device, that was observed at 7000 μmol m$^{-2}$ s$^{-1}$ with this point source LED device.

For the ALE experiment, cells were cultivated in a test tube with 30 mL of modified BG11 medium at 34 °C with aeration under continuous light illumination using a point source LED. Cells were inoculated into fresh medium every 2–3 days to maintain the growth phase. The specific growth rate was calculated from the values obtained at initial and final OD$_{730}$ for each cultivation. Light intensity was set to 7000 μmol m$^{-2}$ s$^{-1}$ at the initial stage of this experiment. After the growth rate increased under a light intensity of 7000 μmol m$^{-2}$ s$^{-1}$, the intensity was gradually increased up to 9000 μmol m$^{-2}$ s$^{-1}$. After the final cultivation, cells which were designated as the Tol strain, were stored at −80 °C with 15% glycerol solution. A frozen stock of the Tol strain obtained by ALE experiment was streaked on a plate containing modified BG11 medium and incubated under a light intensity of 40 μmol m$^{-2}$ s$^{-1}$ at 34 °C using an incubator MIR 154 (Sanyo Electric, Tokyo, Japan). Single colonies thus obtained were inoculated in modified BG11 medium and cultivated at 34 °C with rotary agitation at 150 rpm (BR-43FL, TAITEC, Japan) under continuous light illumination by an LED plate (40 μmol m$^{-2}$ s$^{-1}$; LC-LED 450 W, TAITEC).

**Construction of strains.** All strains used in this study are listed in Supplementary Table 3. A slr1916-deleted strains of the PCC6803 and singly-colony-isolated Tol (S1) strains, designated as 6803Δslr1916 and Tol(S1)Δslr1916, respectively, were constructed by replacing *slr1916* with a chloramphenicol-resistance gene. For constructing the 6803Δslr1916/slr1916m strain, mutated *slr1916* (slr1916m) with its promoter region was amplified from genome DNA of the Tol(S1) strain and was introduced into a neutral site located in *slr0168* of 6803Δslr1916 strain. The hik26-deleted strains of the PCC6803 and Tol(S1) strains, named PCC6803Δhik26 and Tol(S1)Δhik26, were constructed by replacing *hik26* with a kanamycin-resistance gene. For constructing the 6803Δhik26/hik26m strain, mutated *hik26* (hik26m) with its promoter region was amplified from genome DNA of the Tol(S1) strain

and was introduced into a neutral site located at the downstream of the stop codon of *ndhB* in the 6803Δhik26 strain. For constructing the *isiA* overexpressed strain, named 6803/OE-isiA, *isiA* was amplified from genomic DNA of PCC6803. The *isiA* gene with a strong *psbA2* promoter was introduced into a neutral site located in *slr0168* of PCC6803 strain. The procedures for strain constructions are described in Supplementary Methods. The primers used in this study are listed in Supplementary Table 4.

**Analytical methods**. Cell concentration was measured as $OD_{730}$ using a spectrophotometer (UVmini-1240, Shimadzu, Kyoto, Japan). For measuring chlorophyll concentration, 1 mL of culture was centrifuged at 15,000 rpm for 10 min at 4 °C, and 1 mL of 100% methanol was added to the cell pellet. After 15 min, the sample was centrifuged at 15,000 rpm for 10 min at 4 °C, and absorbance of the supernatant thus obtained was measured at 660 nm ($A_{660}$) by a spectrophotometer (UVmini-1240, Shimadzu, Kyoto, Japan). Chlorophyll concentration was calculated by the following formula[39]:

$$\text{Chlorophyll concentration}\left(\mu g\,mL^{-1}\,OD^{-1}\right) = \left(13.4\times A_{665} - 8.3\times A_{650}\right)/OD_{730}$$

For measuring absorption spectra, steady-state fluorescence spectra, and time-resolved fluorescence spectra, cells were grown under light conditions of 4000, 7000, and 9000 μmol m$^{-2}$ s$^{-1}$, and were subsequently collected at 48 h. Next, the samples were dark adapted for 10 min at 34 °C, and centrifuged. Pellets were suspended in 30% polyethylene glycol solution to final concentrations of $OD_{730}$ = 2, and 150 μL of the samples was stored in liquid nitrogen. The absorption spectra of whole cells were measured using a spectrometer (V-650/ISVC-747, JASCO, Tokyo, Japan) at 77 K. Steady-state fluorescence spectra were measured with a spectrofluorometer (FP-6600/PMU-183, JASCO, Tokyo, Japan) at 77 K. The excitation wavelength was 440 nm. Fluorescence decay curves were measured using a time-correlated single-photon counting system at 77-K[40]; the excitation wavelength was 400 nm.

**Whole-genome sequencing**. Total DNA of PCC6803 and tolerant strains were extracted using the Genomic DNA Buffer Set and QIAGEN Genomic-tips 100/G (Qiagen, CA, USA). A DNA library was prepared using Truseq DNA PCR-Free LT sample prep kit (Illumina, CA, USA) and KAPA DNA library quantification kit (KK4824, KAPA Biosystems, MA, USA), and the sample was sequenced using MiSeq sequencer with MiSeq Reagent kit v2 (Illumina) generating 150 bp paired-end reads. The sequence data were mapped to the genome sequence of PCC6803 GT-I strain (NCBI Reference Sequence No.: NC_017038.1)[27] using Bowtie 2 software ver. 2.2.3 with default parameters[41]. SNPs were identified using SAMtools ver 1.0 and BCFtools ver. 1.0[42] with a threshold of a quality score of >150. The sequence data obtained from MiSeq was deposited in DDBJ Sequence Read Archive under accession number DRA010198. The mutations identified by whole-genome sequencing were confirmed by Sanger sequencing using DNA fragments, including the mutation site amplified by PCR using the primer pair listed in Supplementary Table 4.

**DNA microarray**. DNA microarray analysis was performed as described in our previous study with minor modifications[38]. A custom designed microarray of PCC6803 in the Agilent 8 × 60 K platform (Agilent Technologies, Santa Clara, CA) was used. This array contains 10 probes of around 60 nucleotides in length for each open reading frame of PCC6803 genome and its cryptic plasmids, which were designed using a web tool eArray (Agilent Technologies) and genomic sequences of *Synechocystis* sp. PCC6803. The culture broth was mixed with an equal volume of 10% (w/v) phenol in ethanol and then centrifuged to harvest the cells at mid-log growth phase ($OD_{730}$ = around 0.6). The harvested cells were immediately frozen in liquid nitrogen and stored at −80 °C. Total RNA was extracted using Ambion Ribopure yeast kit (Life Technologies Co., Carlsbad, CA). cDNA labeling was performed using Cy3; hybridization and washing were performed using a Low Input Quick Amp WT Labeling Kit (Agilent Technologies), Cyanine 3-CTP (Perkin Elmer), Gene Expression Hybridization Kit (Agilent Technologies), and Gene Expression Wash Buffer (Agilent Technologies). Detection was carried out using an Agilent G2565CA microarray scanner and Feature extraction software (Agilent Technologies). Microarray data were analyzed using Matlab 2014a (Mathworks Inc., Natick, MA). In this study, the median of the intensities of the ten probes representing each ORF was defined as the expression level of the corresponding gene.

**Statistics and reproducibility**. Expression data were normalized by quantile normalization[43] to compare the data to each other. The geometric mean of the median intensity of each gene obtained from triplicate experiments under each condition was used as the representative gene expression level. Gene expression data sets were deposited in Gene Expression Omnibus under accession number GSE149892. The significance of the difference of each gene expression level between the two conditions was analyzed by two-tailed *t*-test. To correct the *p* values for multiple comparisons, the *q* values were calculated by Storey's method[44]. The Bioconducter (version 3.11)[45] with R software (version 4.0.3) was used for the *q* values calculation from *p* values.

**Reporting summary**. Further information on research design is available in the Nature Research Reporting Summary linked to this article.

## Data availability
The whole-genome sequence data were deposited in DDBJ Sequence Read Archive under accession number DRA010198. Gene expression data sets were deposited in Gene Expression Omnibus under accession number GSE149892. All source data for main figures as a Supplementary Data 2.

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

## Acknowledgements

We thank Dr. Yuu Hirose for whole-genome sequencing of the PCC6803 and Tol strains. We also thank Dr. Miyake Chikahiro and Mr. Ginga Shimakawa for their helpful comments. We are grateful to Mr. Tatsuya Itoga for technical support. This work was supported in part by Grants-in-Aid for Scientific Research (Grant Nos. 16H06552 and 16H06559) and by Grants-in-Aid for Challenging Exploratory Research (Grant No. 23656526).

## Author contributions

K.Y., K.O., and H.S. conceived the study. K.Y. and K.O. designed the study and carried out the culture experiments and strain constructions. Y.T., F.M., and H.S. participated in designing the study. S.A. conducted fluorescence analyses. Y.T., K.Y., and H.S. drafted the paper. All the authors contributed toward the preparation of the paper and read and approved the final version of this paper.

## Competing interests

The authors declare no competing interests.
