## [Peer Review File · Communications Biology]

Reviewers' comments:

Reviewer #1 (Remarks to the Author):

The article "Mutations in hik26 and slr1916 lead to high-light stress tolerance in *Synechocystis* sp. PCC6803" describes 2 mutations that lead to a tolerance of photosynthetic cells to very strong light (up to 9000 $\mu\text{mol}/\text{m}^2/\text{s}$). The mutations had been produced through an adaptive laboratory evolution, in which the cells were cultured for 52 days under high light stress conditions. Genome sequencing revealed hik26 and slr1916 as the potential participants in the new route of strong light tolerance.

This is very important and interesting finding, which will be of great interest for academic community specialized in photosynthesis research, as well as for cyanobacterial biotechnologists.

The article is well structured and clearly written, however, some points still require consideration.

1. The deletion of hik26 leads to HL sensitivity, while the T29K mutation leads to HL tolerance. This should be explained. The direct impact of the T29K mutation could be estimated in 6803 Δ hik26 transformed with hik26/T29K. This data is missing in the manuscript.
2. Similar logic may be applied to slr1916/C53Y mutant.
3. It appeared in many publications, that isiAB is induced by various stresses and by a various mutations of regulatory Hiks. At least, isiAB is highly induced by hydrogen peroxide (doi: 10.1111/j.1365-313X.2006.02959.x), which is generated under extremely strong light. Disruption of isiAB resulted in a photosensitive phenotype, with accumulation of reactive oxygen species and cell bleaching in HL (doi: 10.1016/j.febslet.2005.03.021). These facts are well known, and the overexpression experiments of isi in this article seem to be excessive.
4. The expression of slr1926 was linked to the activity of the protein kinase, PmgA - a regulator involved in the HL responses in *Synechocystis* (doi: 10.1038/s41467-020-15491-7). This fact should be mentioned and incorporated into Discussion.
5. slr1928 is known to respond to NaCl stress under control of a Hik34-Rre1 pair. How this is connected to Hik26?
6. A scheme explaining the new mechanism of HL tolerance in Tol mutant would be highly desirable.

Reviewer #2 (Remarks to the Author):

Due to their efficient photosynthesis, Cyanobacteria are considered as promising cell factories for the carbon neutral production of high value chemicals. One limiting step in the production of molecules of interest by cyanobacteria is the photoinhibition in large scale cultures. Therefore it is of a great interest to have a better understanding of mechanism involved in the high-light tolerance. The manuscript from Yoshikawa et al. described the isolation and the characterisation of a high-light tolerant strain of the model cyanobacterium *Synechocystis* sp. PCC 6803 they called Tol. For that purpose an adaptive laboratory evolution (ALE) experiment was performed. To characterise the mutations and the genes involved in the high-light tolerance in this mutant, they used a combination of methods such as: whole genome sequencing, transcriptomic analysis using microarrays, gene deletion, pigments quantification, fluorescence spectra to evaluate photosynthesis capacities.

Interestingly, they identify hik26 as a new player in the photo-tolerance.
Some points need clarification or improvement

- Introduction

Line 46-48: There are only old bibliographic citations (1997-1998). There are more recent publications from many labs concerning the mechanisms involved in photo-inhibition and the players having a role in photo-protection. So may be added some.

Line 50: why 2000 $\mu\text{mole m}^{-2} \text{s}^{-1}$ is considered as the light intensity causing photo-inhibition? There is no citation for this point. In general 300-500 $\mu\text{mole m}^{-2} \text{s}^{-1}$ is used in various laboratories. In their previous work (Ogawa et al 2018 ref N°15) the authors showed that *Synechocystis* PCC 6803 exhibited photo-inhibition when cultured under excessive HL (1100-1300 $\mu\text{mole m}^{-2} \text{s}^{-1}$) they showed that after 24 hours *Synechocystis* PCC 6803 stopped growing when exposed at 1300 $\mu\text{mole m}^{-2} \text{s}^{-1}$. In this study they show that the WT strain is able to grow at 4000 $\mu\text{mole m}^{-2} \text{s}^{-1}$. The authors must discuss more in detail this difference.

Methods and results

- Is the *Synechocystis* strain used in this study non motile? This detail is important because the original wild type of *Synechocystis* is motile. Motile and non motile strains could have a different behaviour.

- Using a ALE experiment is a very good idea to isolate HL tolerant mutant. However, some clarifications could be added concerning the light intensity they used to isolate the Tol mutant. Indeed, they used 7000-9000 $\mu\text{mole m}^{-2} \text{s}^{-1}$ and showed that the WT-type strain is able to grow healthy at 4000 $\mu\text{mole m}^{-2} \text{s}^{-1}$ whereas previously they showed that the WT strain is inhibited at 1300 $\mu\text{mole m}^{-2} \text{s}^{-1}$ (Ogawa et al ref N°15). As they wrote previously, the cell density, the culture tank depth and the wavelength distribution of the light could influence the photo-inhibition rate. So a discussion about the LED spectrum they used could be interesting. The generation time is almost the same when cells are grown under 40 $\mu\text{mole m}^{-2} \text{s}^{-1}$ or 4000 $\mu\text{mole m}^{-2} \text{s}^{-1}$. In general the generation time is higher when cells are grown in low light.

-Fig1. When the photos have been done? After 72h or 48h? There is no indication in the caption. Also missing the legend concerning the white circles (Tol) and dark circles (WT).

Did the Tol mutant is able to grow after 72h at 9000 $\mu\text{mole m}^{-2} \text{s}^{-1}$ or did it stop to grow?

- When chlorophyll content was measured? There is no indication in the caption of figure 3. Probably at 24h or 48h of incubation since the photo of the WT culture grown under 9000 $\mu\text{mole m}^{-2} \text{s}^{-1}$ is 'clear' and no chlorophyll could be detected. The Tol strain looks green-yellow as compared to the wild type when grown at 7000 $\mu\text{mole m}^{-2} \text{s}^{-1}$ Did the authors measured the carotenoids contents?

-Fig.4 Same remark the authors show an absorption spectrum for the WT grown under 4000 $\mu\text{mole m}^{-2} \text{s}^{-1}$ light whereas the photo in fig 1 shows a clear culture.

In the fig.4 the level of chlorophyll in WT strain seems to be similar in all light conditions however the level of phycocyanin and carotenoid is increased. The scale is not the same for the Tol and WT strain so it is difficult to compare the two strain in the panel a.

Is the O₂ production similar in Tol and WT strains?

-Transcriptome analysis. Concerning the supplemental Table 3 have some problem of "format" some parts are not really readable with my computer.

-Concerning the Effect of isiA overexpression on HL tolerance: "Since the isiA expression level in Tol-7000 was higher than that in Tol-4000 and Wild-7000, the enhanced expression of isiA might have

been involved in HL tolerance in the Tol strain ». The authors overexpressed *isiA* in the WT strain, however, the HL tolerance of the PCC6803/OE-*isiA* was lower than that of the Tol strain. Therefore the authors suggested that there are other mechanisms contributing towards HL tolerance of the Tol strain. This is true, however, the authors should have overexpressed the operon *isiA-isiB-sll0249* since these 3 genes are overexpressed in the tol strain (see table 2). *isiB* encodes flavodoxin that replace the photosynthetic ferredoxin when the iron concentration is low.

The genome sequence of the Tol strain harbors two mutations one in *hik26* encoding gene and one in *slr1916*. Therefore the authors constructed knock-out mutants of these gene separately in WT and tol strains . they also introduced the point mutation of *hik26* as well as point mutation . They observed the *hik26* is HL sensitive as compare to the WT showing that the histidine kinase 26 play an important role in the HL stress tolerance.

-Construction of the Δ *hik26/hik26m* strain: the authors wrote that *hik26* (*hik26m*) with its promoter region was amplified from genome DNA of the Tol(S1) strain and was introduced into a neutral site located in *ndhB* of 6803 *hik26* strain". Is there a real neutral site in *ndhB* gene since some known mutants of this gene harbour a phenotype that differs from those of WT?

Is there a reference concerning the pNdhB-psbA2p-EtOH vector40?

The *hik26m* could be introduced at its natural position by introducing the *hik26m* gene in the Δ *hik26::Kmr*.

Reviewer #3 (Remarks to the Author):

Here Yoshikawa and colleagues used laboratory adaptation to generate strains of *Synechocystis* sp. PCC6803 that grew at normal rates in high light levels that inhibited the growth of the parental strain. They assessed levels of photosynthetic pigments, profiled gene expression, and did whole genome sequencing to identify the mutations in an evolved strain. They then constructed new strains to test the roles of the two mutated genes in light tolerance, and confirmed that both genes affect light tolerance. Overall this is a well-performed study with interesting results. There are some issues with data analysis and presentation that should be corrected to make the manuscript suitable for publication.

Statistical comparison of microarray data requires correction for multiple comparisons. Reporting unadjusted p values is not appropriate. Multiple comparisons corrections must be applied before the transcriptomic data can be fully interpreted.

The supplemental materials are confusingly named and presented. The PDF appears to be the same data as in the excel file, but uninterpretable in isolation because of how it's formatted. The PDF should be removed. The excel table should be given a consistent name (probably Supplemental

Table 1) and that name should be used in the text and in the Table heading. In that table, gene expression ratios should be expressed as log₂ ratios, as is the convention.

The other supplemental tables should be presented in a file that indicates their presence (for example, file name "Supplemental tables 2-6").

Throughout, figure legends should contain more information about how the experiment was done, the number and nature of replicates, etc.

I do not have the expertise to evaluate the data shown in figure 4 or the interpretations of these data. Hopefully another reviewer will be able to comment on this section of the manuscript.

I am confused about the phenotype of the parental strain in the 7000 $\mu\text{mol m}^{-2} \text{s}^{-1}$ condition. In the figures in the main text, the growth inhibition appears to be very mild. However, in supplemental figure 1, the growth inhibition phenotype is severe.

Line 272-273: The authors state that "These results indicate that the mutation in hik26 was responsible for the HL tolerance of the Tol strain.". However, deletion of slr1916 also produced heat tolerance. It therefore seems that mutations in either gene can cause heat tolerance, and the phenotype of the Tol strain could be attributed to either or both.

Table 2 should be formatted differently to make it easier to see which data are on which line.

It would be helpful to show the transcriptomic data in volcano plots, to give a broad sense of the number of differentially expressed genes, the magnitude of expression differences, and the directionality of the expression differences.

Lines 286-288, it would be more correct to say that the decrease in chlorophyll content was partially alleviated in the Tol strain (Fig 3 shows a decrease).

The sentence beginning on line 293 that begins "While comparing Wild-4000 to Tol-4000," is unclear and should be reworded. The following sentence is unclear too, in that it describes the data differently from how they are described in the results section. After the authors make the necessary multiple comparisons correction, the number of statistically significantly differentially expressed genes will likely decrease substantially. They will likely need to re-write this and the corresponding section in the results accordingly.

Lines 322-323. Rather than saying genes were probably regulated through Hik26, it would be more appropriate to say that genes "may be regulated" or "could conceivably be regulated". Without looking at their expression in a Hik26 knock out or Hik26 mutant, one cannot attribute expression changes to this protein.

Should the light units be written as " $\mu\text{mol photons m}^{-2} \text{s}^{-1}$ "?

The diagram in Fig 2a is unclear. It would be helpful to draw it such that lines and arrows aren't overlapping each other and culture tubes.

The authors sometimes use "~" where they should use "-" (for example, to show a range, in the

methods and in Fig 2a).

Lines 121-123, it would be clearer to say by what percent chlorophyll content decreased in the Tol strains, rather than saying by what percentage the chlorophyll content decrease was alleviated. Possibly the authors are actually stating the percent decrease in chlorophyll content in the Tol strains, but the wording is confusing.

Line 183, there is a typo – Tol-7000 is listed twice.

Line 182, the clause “and PSI and phycobilisome” is confusing. The sentence should be reworded for clarity.

The authors switch between names for the transcriptomics conditions. For example, Wild-4000 in some places and W-4000 in others. They should choose a format and use it consistently.

Responses to Reviewer #1

We would like to thank to the reviewer for his/her insightful comments. All revised parts have been red-color font in the manuscript.

1. The deletion of *hik26* leads to HL sensitivity, while the T29K mutation leads to HL tolerance. This should be explained. The direct impact of the T29K mutation could be estimated in *6803Δhik26* transformed with *hik26/T29K*. This data is missing in the manuscript.

Response: We thank the reviewer for the appropriate comment. The label of the original Fig. 5i was not Tol(S1) Δ *hik26*/*hik26m* but *6803Δhik26*/*hik26m*. The growth profiles of parent (PCC6803), Tol, *6803Δhik26*, and *6803Δhik26*/*hik26m* are shown in below.

In the manuscript, the growth character of *6803Δhik26*/*hik26m* was described as the following sentence.

Lines 271–273:

The *6803Δhik26*/*hik26m* showed identical growth as that of the Tol strain under all conditions, even under $9000 \mu\text{mol m}^{-2} \text{s}^{-1}$ (Fig. 5j); however, high viscosity and cell aggregation were not observed in the *6803Δhik26*/*hik26m*.

These results indicate that the *hik26* mutation (T29K) was responsible for the HL tolerance of the Tol strain.

2. Similar logic may be applied to *slr1916*/*C53Y* mutant.

Response: To address the reviewer's concerns, we constructed the *6803Δslr1916*/*slr1916m* introduced the *slr1916* mutation into the neutral site (*slr0168*) of the *6803Δslr1916*. The cell growths under the various light conditions were added to the revised Fig. 5g. The growth profiles of parent, Tol, and *6803Δslr1916*/*slr1916m* are shown in below.

The 6803 Δ slr1916/slr1916m showed almost identical growth as that of the Tol and the 6803 Δ slr1916 strains under all conditions. Furthermore, the culture broth of PCC6803 Δ slr1916/slr1916m showed high viscosity and cell aggregation under the HL conditions, as well as PCC6803 Δ slr1916. Following sentences were added to the revised manuscript.

Lines 280–286:

As well as *hik26*, mutated *slr1916* (hereafter called as *slr1916m*) with its own promoter region was amplified from the Tol(S1) strain and introduced into the genome of the 6803 Δ slr1916, generating 6803 Δ slr1916/slr1916m. The 6803 Δ slr1916/slr1916m showed almost identical growth as that of the Tol and the 6803 Δ slr1916 strains under all conditions (Fig. 5g). Furthermore, the culture broth of PCC6803 Δ slr1916/slr1916m showed high viscosity and cell aggregation under the HL conditions, as well as PCC6803 Δ slr1916. These results suggest that the mutation of *slr1916* in the Tol strain is loss-of-function mutation.

3. It appeared in many publications, that *isiAB* is induced by various stresses and by a various mutations of regulatory Hiks. At least, *isiAB* is highly induced by hydrogen peroxide (doi: 10.1111/j.1365-313X.2006.02959.x), which is generated under extremely strong light. Disruption of *isiAB* resulted in a photosensitive phenotype, with accumulation of reactive oxygen species and cell bleaching in HL (doi: 10.1016/j.febslet.2005.03.021). These facts are well known, and the overexpression experiments of *isi* in this article seem to be excessive.

Response: We thank the reviewer for providing important publications about *isiAB*. Although it may be possible to predict the phenotype by *isiA* overexpression from previous knowledges with superior insight, the experimental data was needed to interpret and support the transcriptome data. According to the reviewer's suggestion, we simplified the description of *isiA* overexpression experiment and added the previously known findings in the discussion

(Lines 308-310).

4. *The expression of slr1926 was linked to the activity of the protein kinase, PmgA - a regulator involved in the HL responses in Synechocystis (doi: 10.1038/s41467-020-15491-7). This fact should be mentioned and incorporated into Discussion.*

Response: We thank the reviewer for providing the important publication about regulation of *slr1916* expression. It will be a clue for our future study to reveal the regulatory mechanisms involves in the high light tolerance of the Tol strain. Following discussion was added to the revised manuscript.

Lines 339–343:

Regarding the *slr1916*, recently, comprehensive experiments using an inducible CRISPRi gene repression library have shown that *pmgA* and *slr1916* repression clones exhibit similar growth profiles in response various stresses including light conditions (Yao et al., 2020). The PgmA is a regulator involved in HL responses in the PCC6803 (Hihara et al. 1998). The expression of *slr1916* would be linked to the PgmA activity.

5. *slr1928 is known to respond to NaCl stress under control of a Hik34-Rre1 pair. How this is connected to Hik26?*

Response: As reviewer suggested, identification of a pair of histidine kinase and response regulator is very important to reveal the mechanism of two component system in response to signal. In PCC6803, many histidine kinases and response regulators have been investigated under various stress environments. Identification of a response regulator of the Hik26 would reveal the relationship to the other two component systems. We would like to clarify these in our future research.

6. *A scheme explaining the new mechanism of HL tolerance in Tol mutant would be highly desirable.*

Response: According to the reviewer's suggestion, a scheme of the mechanism of HL tolerance in the Tol was added to the discussion as Fig. 6.

Fig. 6 A scheme of the mechanism of HL tolerance in the Tol strain. The Hik26 is a histidine kinase in response to HL stress. The mutation (T29K) of Hik26 in the Tol strain may activate the histidine kinase activity. The Hik26 induces the expression of genes including *isiA* through a response regulator. The *IsiA* contributes to protect the membrane proteins from the oxidative stress caused by HL conditions, and involves in light state transition from PSII to PSI.

Response to Reviewer #2

We would like to thank to the reviewer for his/her insightful comments. All revised parts have been red-color font in the manuscript.

1. Line 46-48: *There are only old bibliographic citations (1997-1998). There are more recent publications from many labs concerning the mechanisms involved in photo-inhibition and the players having a role in photo-protection. So may be added some.*

Response: According to the reviewer's suggestion, we revised the sentence with citations of recent publications as below.

Lines 41-44:

Furthermore, they possess some mechanisms for preventing photoinhibition by allowing excess light energy to escape, such as state transition (Campbell et al., 1998), heat dissipation (Niyogi, 1999), energy dissipation from phycobilisome and photosystem II (PSII) (Ruban et al., 2007; Ivanov et al., 2008), and synthesis and maintenance of protein complexes in thylakoid membrane by chaperone proteins (Thurotte et al., 2020).

2. Line 50: why 2000 $\mu\text{mole m}^{-2} \text{s}^{-1}$ is considered as the light intensity causing photo-inhibition? There is no citation for this point. In general 300-500 $\mu\text{mole m}^{-2} \text{s}^{-1}$ is used in various laboratories. In their previous work (Ogawa et al 2018 ref N°15) the authors showed that *Synechocystis* PCC 6803 exhibited photo-inhibition when cultured under excessive HL (1100-1300 $\mu\text{mole m}^{-2} \text{s}^{-1}$) they showed that after 24 hours *Synechocystis* PCC 6803 stopped growing when exposed at 1300 $\mu\text{mole m}^{-2} \text{s}^{-1}$. In this study they show that the WT strain is able to grow at 4000 $\mu\text{mole m}^{-2} \text{s}^{-1}$. The authors must discuss more in detail this difference.

Response: This difference comes from the light device and the culture vessels. In our previous paper (Ogawa et al., 2018), we used a flat LED panel with the photobioreactor. However, in the present study, we culture the cells in a test-tube with a point source LED that can irradiate more high light intensity. The light intensity was measured at the irradiate point on the surface of the test-tube. Therefore, the normalized light intensity at the entire test-tube becomes low. In the PCC6803 strain, although 70% growth reduction was observed at 1100 $\mu\text{mol m}^{-2} \text{s}^{-1}$ with the flat LED device, that was observed at 7000 $\mu\text{mol m}^{-2} \text{s}^{-1}$ with this point source LED device. Following explanations were added to the revised manuscript.

Lines 371–378:

In our previous report (Ogawa et al., 2018), the PCC6803 strain exhibited photo-inhibition when cultured under 1100 ~ 1300 $\mu\text{mol m}^{-2} \text{s}^{-1}$. Although a flat LED panel with the photobioreactor was used in the previous study, we here culture the cells in a test-tube with a point source LED that can irradiate more high light intensity. The light intensity was measured at the irradiate point on the surface of the test-tube. Therefore, the normalized light intensity at the entire test-tube becomes low. In the PCC6803 strain, although 70% growth reduction was observed at 1100 $\mu\text{mol m}^{-2} \text{s}^{-1}$ with the flat LED device, that was observed at 7000 $\mu\text{mol m}^{-2} \text{s}^{-1}$ with this point source LED device.

3. Is the *Synechocystis* strain used in this study non motile? This detail is important because the original wild type of *Synechocystis* is motile. Motile and non motile strains could have a different behaviour.

Response: In the present study, we used *Synechocystis* sp. PCC6803 GT-I strain as a parental strain. It has been reported that the GT-I strain is non motile (Trautmann et al., 2012). The information was added to the lines 345-346 in the revised manuscript.

Trautmann D, Voss B, Wilde A, Al-Babili S, Hess WR. Microevolution in cyanobacteria:

re-sequencing a motile substrain of *Synechocystis* sp. PCC 6803. DNA Res 19, 435-448 (2012).

4. Using a ALE experiment is a very good idea to isolate HL tolerant mutant. However, some clarifications could be added concerning the light intensity they used to isolate the Tol mutant. Indeed, they used 7000-9000 $\mu\text{mole m}^{-2} \text{s}^{-1}$ and showed that the WT-type strain is able to grow healthy at 4000 $\mu\text{mole m}^{-2} \text{s}^{-1}$ whereas previously they showed that the WT strain is inhibited at 1300 $\mu\text{mole m}^{-2} \text{s}^{-1}$ (Ogawa et al ref N°15). As they wrote previously, the cell density, the culture tank depth and the wavelength distribution of the light could influence the photo-inhibition rate. So a discussion about the LED spectrum they used could be interesting. The generation time is almost the same when cells are grown under 40 $\mu\text{mole m}^{-2} \text{s}^{-1}$ or 4000 $\mu\text{mole m}^{-2} \text{s}^{-1}$. In general the generation time is higher when cells are grown in low light.

Response: As the response to the comment 2, this difference comes from the light device and the culture vessels. We used a flat LED panel in the previous study, and used a point source LED in the present study. Since the light intensity was measured at the irradiate point on the surface of the test-tube, the normalized light intensity at the entire test-tube becomes low. The explanations were added to the lines 369-376 in revised manuscript.

Regarding the light spectrum, we used white LED as a point light source. The spectrum was added to the supplementary Fig. S3.

Supplementary Fig. S3 Spectral profile of the point source LED light.

Total light intensity was adjusted to 1000 $\mu\text{mol m}^{-2} \text{s}^{-1}$ light. The spectral profile was measured with a light analyzer (LA-105, NK Systems, Japan).

The culture with 40 $\mu\text{mol m}^{-2} \text{s}^{-1}$ light was performed in a flask scale with not the point source

LED but the flat LED panel. As reviewer mentioned, the growth rate increases with increasing light intensity. But, when the light intensity exceeds a certain level, the growth rate becomes constant. The difference of light source for $40 \mu\text{mol m}^{-2} \text{s}^{-1}$ light was noted to the legends of Figs. 1, and 5.

5. *Fig1. When the photos have been done? After 72h or 48h? There is no indication in the caption. Also missing the legend concerning the white circles (Tol) and dark circles (WT). Did the Tol mutant is able to grow after 72h at $9000 \mu\text{mole m}^{-2} \text{s}^{-1}$ or did it stop to grow?*

Response: We are sorry for lacking the information. The photos in Fig. 1 were obtained at 72 h. The information was added to the figure legend. Because the cultures were stopped at 72 h, we do not have the growth data after 72 h.

6. *When chlorophyll content was measured? There is no indication in the caption of figure 3. Probably at 24h or 48h of incubation since the photo of the WT culture grown under $9000 \mu\text{mole m}^{-2} \text{s}^{-1}$ is 'clear' and no chlorophyll could be detected. The Tol strain looks green-yellow as compared to the wild type when grown at $7000 \mu\text{mole m}^{-2} \text{s}^{-1}$ Did the authors measured the carotenoids contents?*

Response: We are sorry for lacking the information. The chlorophyll contents were measured at 72 h. The information was added to the legend of Fig. 3. We did not measure the carotenoids contents, but the increase of carotenoids contents in the Tol strain were not supported by the absorption spectra in Fig. 4.

7. *Fig.4 Same remark the authors show an absorption spectrum for the WT grown under $4000 \mu\text{mole m}^{-2} \text{s}^{-1}$ light whereas the photo in fig 1 shows a clear culture. In the fig.4 the level of chlorophyll in WT strain seems to be similar in all light conditions however the level of phycocyanin and carotenoid is increased. The scale is not the same for the Tol and WT strain so it is difficult to compare the two strain in the panel a. Is the O_2 production similar in Tol and WT strains?*

Response: According to the reviewer's suggestion, the same scale was used for the PCC6803 (WT) and Tol strains in the Fig. 4a. The absorption spectra show that the relative content of chlorophyll (675 nm) to phycobilisome (620 nm) decreased as the light intensity increased in the WT strain, it was stable in the Tol strain. As the reviewer pointed, the comparison suggests increase of the relative carotenoid content in the WT was larger than that in the Tol strain. We did not measure the oxygen production rates. This discussion was added to the lines 133-134 of

the revised manuscript.

8. *Transcriptome analysis. Concerning the supplemental Table 3 have some problem of “format” some parts are not really readable with my computer.*

Response: We apologize for the invisibility of the supplemental Table 3. The format of the table was revised to be more readable.

9. *Concerning the Effect of isiA overexpression on HL tolerance: “Since the isiA expression level in Tol-7000 was higher than that in Tol-4000 and Wild-7000, the enhanced expression of isiA might have been involved in HL tolerance in the Tol strain ». The authors overexpressed isiA in the WT strain, however, the HL tolerance of the PCC6803/OE-isiA was lower than that of the Tol strain. Therefore the authors suggested that there are other mechanisms contributing towards HL tolerance of the Tol strain. This is true, however, the authors should have overexpressed the operon isiA-isiB-sll0249 since these 3 genes are overexpressed in the tol strain (see table 2). isiB encodes flavodoxin that replace the photosynthetic ferredoxin when the iron. concentration is low.*

Response: We thank the reviewer for the constructive comment. Following discussion was added to the revised manuscript.

Lines 228–230:

Furthermore, because the *isiA-isiB-sll0249* is an operon on the genome, the expressions of *isiB* and *sll0249* as well as *isiA* were increased in the Tol strain (Table 2). The overexpression of these three genes may lead to stronger HL tolerance.

10. *Contraction of the Δ hik26/hik26m strain: the authors wrote that hik26 (hik26m) with its promoter region was amplified from genome DNA of the Tol(S1) strain and was introduced into a neutral site located in ndhB of 6803□hik26 strain”. Is ther a real neutral site in ndhB gene since some known mutants of this gene harbour a phenotype that differs from those of WT?*

Response: According to the following reference (Takahashi et al., 2010), we considered the downstream region of the *ndhB* gene as the neutral site. Indeed, as the reviewer mentioned at the Comment 12, it would be better to introduce the mutation in its original position on the genome.

Takahashi T, Nakai N, Muramatsu M, Hihara Y. Role of multiple HLR1 sequences in the

regulation of the dual promoters of the *psaAB* genes in *Synechocystis* sp. PCC 6803. *J Bacteriol* 192, 4031-4036 (2010).

11. Is there a reference concerning the *pNdhB-psbA2p-EtOH* vector40?

Response: We are sorry for missing the information. The list of references was added to the supplementary file. The vector was derived from following reference.

Yoshikawa K, Hirasawa T, Shimizu H. Effect of malic enzyme on ethanol production by *Synechocystis* sp. PCC 6803. *J Biosci Bioeng* 119, 82-84 (2015).

12. The *hik26m* could be introduced at its natural position by introducing the *hik26m* gene in the *Δhik26::Kmr*.

Response: We thank the reviewer for suggesting the reverse engineering method. In our future research, we will return mutations to its original position in on the genome.

Response to Reviewer #3

We would like to thank to the reviewer for his/her insightful comments. All revised parts have been red-color font in the manuscript.

1. *Statistical comparison of microarray data requires correction for multiple comparisons. Reporting unadjusted p values is not appropriate. Multiple comparisons corrections must be applied before the transcriptomic data can be fully interpreted.*

Response: According to the reviewer's suggestion, the *p* values were corrected to the *q* values for considering false discoveries by Storey's method (Storey and Tibshirani, 2003). The Bioconductor (version 3.11) with R software (version 4.0.3) was used for the calculation.

As the result using *q*-values for statistic evaluation, the numbers of genes whose expression levels are changed in Table 1 were decreased.

Table 1. Number of genes showing significant difference in expression levels.

	W-4000	W-7000	T-4000	T-7000
W-4000	–	2 (0.1%)	13 (0.4%)	455 (15.5%)

W-7000	–	–	147 (5.0%)	824 (28.0%)
T-4000	–	–	–	2 (0.1%)
T-7000	–	–	–	–

Corresponding parts were revised as below in the revised manuscript.

Lines 165–172:

Differences were compared with the two tailed Student's t-test. To correct the p values for multiple comparisons, the q values were calculated with considering false discoveries by Storey's method. When the expression data between 4000 and 7000 $\mu\text{mol m}^{-2} \text{s}^{-1}$ were compared within each condition, only a small number of genes showed different expression levels in both the strains (0.1% between W-4000 and W-7000; 0.1% between T-4000 and T-7000). This result suggested that the 4000 $\mu\text{mol m}^{-2} \text{s}^{-1}$ condition for the Tol strain is also an HL condition for PCC6803. However, after comparing the expression data between PCC6803 and Tol strains, a large number of genes (28.0%) showed different expression levels under 7000 $\mu\text{mol m}^{-2} \text{s}^{-1}$.

2. The supplemental materials are confusingly named and presented. The PDF appears to be the same data as in the excel file, but uninterpretable in isolation because of how it's formatted. The PDF should be removed. The excel table should be given a consistent name (probably Supplemental Table 1) and that name should be used in the text and in the Table heading. In that table, gene expression ratios should be expressed as log2 ratios, as is the convention.

Response: We apologize for the invisibility of the gene expression data. According to the reviewer's suggestion, the log2 ratios were summarized in Supplemental Table 1 as a excel file with consistent file name. The PDF file was removed.

3. The other supplemental tables should be presented in a file that indicates their presence (for example, file name "Supplemental tables 2-6"). Throughout, figure legends should contain more information about how the experiment was done, the number and nature of replicates, etc.

Response: According to the reviewer's suggestion, we extracted the Supplemental tables 2-6, and presented a file name of "Supplemental_Table_2-6.xlsx". We carefully checked the figure legends to satisfy the required information.

4. I am confused about the phenotype of the parental strain in the 7000 $\mu\text{mol m}^{-2} \text{s}^{-1}$ condition. In the figures in the main text, the growth inhibition appears to be very mild. However, in supplemental figure 1, the growth inhibition phenotype is severe.

Response: We are sorry for confusing the reviewer. The growth curve of the supplemental Fig. 1 was not 7000 $\mu\text{mol m}^{-2} \text{s}^{-1}$ but 9000 $\mu\text{mol m}^{-2} \text{s}^{-1}$. The growth profiles were almost identical between Fig. 1 and Fig. S1.

5. Line 272-273: The authors state that “These results indicate that the mutation in *hik26* was responsible for the HL tolerance of the Tol strain. “. However, deletion of *slr1916* also produced heat tolerance. It therefore seems that mutations in either gene can cause heat tolerance, and the phenotype of the Tol strain could be attributed to either or both.

Response: We thank the reviewer for suggesting the appropriate description. The sentence was revised as below.

Lines 273–275:

Both the *hik26* mutation and *slr1916* deletion also produced HL tolerance. Therefore, it seems that the mutations in either gene can cause HL tolerance, and the phenotype of the Tol strain could be attributed to either or both.

6. Table 2 should be formatted differently to make it easier to see which data are on which line.

Response: The format of Table 2 was revised to make it more readable.

7. It would be helpful to show the transcriptomic data in volcano plots, to give a broad sense of the number of differentially expressed genes, the magnitude of expression differences, and the directionality of the expression differences.

Response: According to the reviewer’s suggestion, the comparison between T-7000 and W-7000 is shown as volcano plot. The increase of *isiA* and *isiB* expressions at T-7000 are clearly shown in this graph.

Supplementary Fig. S4 Volcano plot for finding genes whose expressions were significantly changed in terms of two-tailed Student's *t*-test ($\alpha < 0.05$) and the fold change (> 5 and < 0.2).

8. Lines 286-288, it would be more correct to say that the decrease in chlorophyll content was partially alleviated in the Tol strain (Fig 3 shows a decrease).

Response: According to the reviewer suggestion, we revised the sentence.

9. The sentence beginning on line 293 that begins “While comparing Wild-4000 to Tol-4000,” is unclear and should be reworded. The following sentence is unclear too, in that it describes the data differently from how they are described in the results section. After the authors make the necessary multiple comparisons correction, the number of statistically significantly differentially expressed genes will likely decrease substantially. They will likely need to re-write this and the corresponding section in the results accordingly.

Response: As reviewer suggested, the number of statistically significantly differentially expressed genes decreased by the correction for multiple comparisons. The part was revised as below.

Lines 301–303:

Although a comparison between W-4000 and T-4000 showed that the number of genes whose expression was significantly changed (0.4%), a comparison between W-7000 and T-7000 showed that the expression levels of many genes were changed (28.0%).

10. Lines 322-323. Rather than saying genes were probably regulated through Hik26, it would be more appropriate to say that genes “may be regulated” or “could conceivably be regulated”. Without looking at their expression in a Hik26 knock out or Hik26 mutant, one cannot attribute expression changes to this protein.

Response: We thank the reviewer for correcting the sentence. We changed the “were probably regulated” to “may be regulated”.

11. Should the light units be written as “ $\mu\text{mol photons m}^{-2} \text{s}^{-1}$ ”?

Response: We revised all light units as “ $\mu \text{ mol photons m}^{-2} \text{ s}^{-1}$ ”.

12. The diagram in Fig 2a is unclear. It would be helpful to draw it such that lines and arrows aren't overlapping each other and culture tubes.

Response: The Fig. 2a was revised so that the lines and arrows are not overlapping each other and culture tubes.

Revised Fig. 2a ALE experiment under high light stress condition. a. Experimental design of the ALE experiment. Cells were cultured in a test tube using a point source LED. Among four parallel cultures, the culture broth with the highest growth rate was inoculated into a fresh medium every two to three days.

13. The authors sometimes use “~” where they should use “-” (for example, to show a range, in the methods and in Fig 2a).

Response: We changed “~” to “-” throughout the manuscript, figures, and supplementary files.

14. Lines 121-123, it would be clearer to say by what percent chlorophyll content decreased in the Tol strains, rather than saying by what percentage the chlorophyll content decrease was alleviated. Possibly the authors are actually stating the percent decrease in chlorophyll content in the Tol strains, but the wording is confusing.

Response: We revised the description of this sentence as below.

Lines 118–120:

In the Tol strain, the 30% and 22% of the decreased chlorophyll content were alleviated under 7000 and 9000 $\mu\text{mol m}^{-2} \text{s}^{-1}$, respectively.

15. Line 183, there is a typo – Tol-7000 is listed twice. Line 182, the clause “and PSI and phycobilisome” is confusing. The sentence should be reworded for clarity.

Response: Thank you very much for pointing out the unreadable part. The sentences were revised as below.

Lines 181–183:

No overrepresented functional category was found between W-4000 and W-7000. The comparison between T-4000 and T-7000 suggests the expression of genes such as *psa*, *cpc*, and *apc*, coding proteins for PSI and phycobilisome, was low in Tol-4000.

16. The authors switch between names for the transcriptomics conditions. For example, Wild-4000 in some places and W-4000 in others. They should choose a format and use it consistently.

Response: We thank the reviewer for pointing out the usage. We used the appropriate names for the transcriptomics conditions to make the manuscript consistent.

REVIEWERS' COMMENTS:

Reviewer #1 (Remarks to the Author):

The article was appropriately revised. I believe it is now acceptable for publication. Some minor English proofreading of newly added sentences is desirable.

Reviewer #2 (Remarks to the Author):

The authors dealt well with the referees comments and really improve the manuscript. Nevertheless I have already some comments.

Point 5 reviewer 2 : It is very easy to perform growth curves (1-2 weeks). So to my opinion it is very important to show that the mutant continue to be more tolerant to High light as compared to the wild-type after 72h. May be the wild type need an adaptation time to be able to growth at 9000 $\mu\text{mol}/\text{m}^2/\text{s}$. If it is so, the interpretation is that the mutant does not need adaptation.

Point 6 As shown in figure 4 the carotenoids contents of the mutant is decreased and not increased compared to the wild-type.

Point 12: I continue to think that The hik26m could be introduced at its natural position by introducing the hik26m gene in the $\Delta\text{hik26}::\text{Kmr}$. and not in a future publication.

Reviewer #3 (Remarks to the Author):

This revised manuscript is much improved. However, there are still some additional corrections and clarifications needed, as listed below. Additionally, the manuscript would benefit from English language editing by someone fluent in English grammar.

Fig S2c, the labels for PCC6803 and Tol are swapped.

Lines 118-120: although this sentence was re-written, it is still unclear. I suggest changing it to something like: "In the Tol strain, chlorophyll content was only decreased by x% and y% under 7000 and 9000 $\mu\text{mol m}^{-2} \text{s}^{-1}$, respectively."

Lines 168-170: The wording is confusing. I suggest changing it to something like "When the expression data between 4000 and 7000 $\mu\text{mol m}^{-2} \text{s}^{-1}$ were compared for each strain, only a small number of genes showed different expression levels ((0.1% between W-4000 and W-7000; 0.1% between T-4000 and T-7000)."

Lines 170-172: The wording is confusing. I don't understand the point that the authors are trying to make.

Lines 180-186: This paragraph and Supplemental Table 2 refer to analysis done before the multiple comparisons correction. This analysis must be redone using only genes with significant q values.

Since only 2 genes differed significantly between 4000 and 7000 $\mu\text{mol}/\text{m}^2/\text{s}$ for each strain, enrichment analysis will only be possible in comparisons between strains.

Supplemental Table 2: see above.

Figure S4: the significance threshold should be set using the q values rather than the p values.

Table 2: this table should be revised to include only genes with significant q values, or if the authors wish to use the current lists they should include the q values in the table so that readers can weight the significance accordingly.

Lines 204-206: The authors should include a qualifying statement that the differences between T-4000 and T-7000 for Tol were not statistically significant.

Lines 303-306: The new sentence added in red seems to be missing words. Perhaps the authors meant to say "...whose expression was significantly changed was small (0.4%), a comparison..."

Lines 345-346: I don't understand what is meant by "The expression of slr1916 would be linked to the PgmA activity." Was it linked in the cited paper? Or the authors are hypothesizing that it could possibly be linked?

Strain construction as described in the methods section and table S5: when the authors say a neutral site is located IN a gene, do they mean within the coding sequence (which could conceivably be neutral if the gene is a pseudogene) or downstream of the stop codon? NS1 is stated as being in the ndhB gene in the methods but downstream of ndhB in table S5.

Response to Reviewer #1

We would like to thank to the reviewer for his/her positive evaluation. We checked the English grammar of newly added sentences.

Response to Reviewer #2

We would like to thank to the reviewer for his/her insightful comments. Regarding the chlorophyll contents in Fig. 3, although we answered that the data was obtained at 72 h in the previous response, we used the data at 48 h not at 72 h for this graph. We sincerely apologize the error and now revised the information in the figure legend. All revised parts have been red-color font in the manuscript.

1. Point 5: It is very easy to perform growth curves (1-2 weeks). So to my opinion it is very important to show that the mutant continue to be more tolerant to High light as compared to the wild-type after 72h. May be the wild type need an adaptation time to be able to growth at 9000 $\mu\text{mol}/\text{m}^2/\text{s}$. If it is so, the interpretation is that the mutant does not need adaptation.

Response: We understand the meaning of your comment. The growth curves until 96 h of wild type and Tol strains under 9000 $\mu\text{mol m}^{-2} \text{s}^{-1}$ are shown in the Supplementary Fig. 1 (Please see the red symbols). No further growth of wild type strain was observed after 72 h under the high light condition. So, following sentence was added to the legend of Fig. 1.

Lines 542 – 545:

No further growth was observed in the PCC6803 strain after 72 h under the 9000 $\mu\text{mol m}^{-2} \text{s}^{-1}$ (Supplementary Fig. 1).

Supplementary Fig. 1 Evaluation of HL tolerance stability of the Tol strain. The PCC6803 and Tol strains were cultured under 4000 and 9000 $\mu\text{mol m}^{-2} \text{s}^{-1}$ after with or without 15 days subcultures under low light condition (40 $\mu\text{mol m}^{-2} \text{s}^{-1}$). The panels of a and b, and c and d represent the PCC6803 and Tol strains, respectively. The left and right panels show growth under 4000 and 9000 $\mu\text{mol m}^{-2} \text{s}^{-1}$. Error bars indicate standard deviation of triplicate cultures.

2. Point 6 *As shown in figure 4 the carotenoids contents of the mutant is decreased and not increased compared to the wild-type.*

Response: Thank you very much for carefully reading our manuscript. The corresponding sentence was revised as below.

Lines 117 – 118:

The comparison of absorption spectra suggests the relative carotenoid content in the Tol strain is decreased compared to PCC6803 strain.

3. Point 12: *I continue to think that The hik26m could be introduced at its natural position by introducing the hik26m gene in the $\Delta\text{hik26}::\text{Kmr}$. and not in a future publication.*

Response: We thank the reviewer for this comment. To note the potential issue to readers, following sentence was added to the legend of Fig. 5, as below.

Lines 572 – 574:

Note that, in the 6803 $\Delta\text{slr1916}/\text{slr1916m}$ and 6803 $\Delta\text{hik26}/\text{hik26m}$ strains, the mutated genes were introduced at the neutral sites, not at their natural positions.

Response to Reviewer #3

We would like to thank to the reviewer for his/her insightful comments. We carefully checked the English grammar and edited the manuscript. All revised parts have been red-color font in the manuscript.

1. Fig S2c, *the labels for PCC6803 and Tol are swapped.*

Response: We thank the reviewer for the careful reading of our manuscript. The pointed error of labels was revised in the Supplementary file.

2. Lines 118-120: although this sentence was re-written, it is still unclear. I suggest changing it to something like: “In the Tol strain, chlorophyll content was only decreased by x% and y% under 7000 and 9000 $\mu\text{mol m}^{-2} \text{s}^{-1}$, respectively.”

Response: According to the reviewer’s suggestion, we revised this sentence as below.

Lines 108 – 109:

In the Tol strain, chlorophyll content was only decreased by 21% and 42% under 7000 and 9000 $\mu\text{mol m}^{-2} \text{s}^{-1}$, respectively.

3. Lines 168-170: The wording is confusing. I suggest changing it to something like “When the expression data between 4000 and 7000 $\mu\text{mol m}^{-2} \text{s}^{-1}$ were compared for each strain, only a small number of genes showed different expression levels ((0.1% between W-4000 and W-7000; 0.1% between T-4000 and T-7000)).

Response: We revised the sentence according the reviewer’s suggestion.

4. Lines 170-172: The wording is confusing. I don’t understand the point that the authors are trying to make.

Response: We thank the reviewer for pointing out the sentence should be revised. This sentence was re-written as below.

Lines 147 – 149:

In the comparison between W-7000 and T-7000, the expression levels of 28% genes shows significant difference between PCC6803 and Tol strains under 7000 $\mu\text{mol m}^{-2} \text{s}^{-1}$.

5. Lines 180-186: This paragraph and Supplemental Table 2 refer to analysis done before the multiple comparisons correction. This analysis must be redone using only genes with significant q values. Since only 2 genes differed significantly between 4000 and 7000 $\mu\text{mol/m}^2/\text{s}$ for each strain, enrichment analysis will only be possible in comparisons between strains.
Supplemental Table 2: see above.

Response: As the reviewer suggested, the functional enrichment analysis should be performed using only genes with significant q-value. On the reconsideration, we decided to remove the paragraph regarding the enrichment analysis with the Supplementary Table 2, because the

central finding obtained from the transcriptome data was the expression levels of *isiA* and *isiB* were dramatically increased in the Tol strain under the HL condition as shown in Table 2.

6. *Figure S4: the significance threshold should be set using the q values rather than the p values.*

Response: According to the reviewer's suggestion, the *q*-values were used for the volcano plot.

7. *Table 2: this table should be revised to include only genes with significant q values, or if the authors wish to use the current lists they should include the q values in the table so that readers can weight the significance accordingly.*

Response: We thank the reviewer for this suggestion. Only genes with significant *q*-values were considered for the ranking. The column of *q*-value was added to the Table 2. Only *ss15070* and *sll1514* were removed from the table, but the difference does not affect the story of this manuscript.

8. *Lines 204-206: The authors should include a qualifying statement that the differences between T-4000 and T-7000 for Tol were not statistically significant.*

Response: As pointed by the reviewer, no statistical significance was shown in *isiA* expression levels between T-4000 and T-7000. Therefore, this sentence was removed from the manuscript.

9. *Lines 303-306: The new sentence added in red seems to be missing words. Perhaps the authors meant to say "...whose expression was significantly changed was small (0.4%), a comparison..."*

Response: We thank the reviewer for pointing out the missing words. This sentence was revised in accordance with the reviewer's suggestion.

10. *Lines 345-346: I don't understand what is meant by "The expression of *slr1916* would be linked to the *PgmA* activity." Was it linked in the cited paper? Or the authors are hypothesizing that it could possibly be linked?*

Response: We have no experimental evidence to connect the *slr1916* expression and *PgmA* activity. Only similar growth profiles of the *pmgA* and *slr1916* repression clones are shown in the cited paper. So, this sentence was removed from the manuscript.

11. Strain construction as described in the methods section and table S5: when the authors say a neutral site is located IN a gene, do they mean within the coding sequence (which could conceivably be neutral if the gene is a pseudogene) or downstream of the stop codon? NS1 is stated as being in the *ndhB* gene in the methods but downstream of *ndhB* in table S5.

Response: The location of NS1 is downstream of the stop codon of *ndhB* as described in Table S5. The description in the methods section was revised as below.

Lines 334 – 335:

“... was introduced into a neutral site located at the downstream of the stop codon of *ndhB* in the 6803*Ahik26* strain.”